# Deep Learning Radiographic Assessment of Pulmonary Edema: Training with Serum Biomarkers

**Justin Huynh**[1]      JUH079@UCSD.EDU
[1] *University of California San Diego, San Diego, CA, USA*

**Samira Masoudi**[1]      SMASOUDI@UCSD.EDU
**Abraham Noorbakhsh**[1]      ANOORBAK@UCSD.EDU
**Amin Mahmoodi**[1]      AMAHMOODI@HEALTH.UCSD.EDU
**Kyle Hasenstab**[2]      KAHASENSTAB@SDSU.EDU
[2] *San Diego State University, San Diego, California, USA*

**Michael Pazzani**[1]      MPAZZANI@UCSD.EDU
**Albert Hsiao**[1]      A3HSIAO@HEALTH.UCSD.EDU

**Editors:** Under Review for MIDL 2022

## Abstract

A major obstacle faced when developing convolutional neural networks (CNNs) for medical imaging is the acquisition of training labels: most current approaches rely on manually prescribed labels from physicians, which are time consuming and labor intensive to attain. Clinical biomarkers, often measured alongside medical images and used in diagnostic workup, may provide a rich set of data that can be collected retrospectively and utilized to train diagnostic models. In this work, we focused on the blood serum biomarkers BNP and BNPP, indicative of acute heart failure (HF) and cardiogenic pulmonary edema, paired with the chest X-ray imaging modality. We investigated the potential for inferring BNP and BNPP from chest radiographs. For this purpose, a CNN was trained using 27748 radiographs to automatically infer BNP and BNPP, and achieved strong performance ($AUC = 0.90$, $SEN = 0.88$, $SPEC = 0.81$, $r = 0.79$). Since radiographic features of pulmonary edema may not be visible on low resolution images, we also assessed the impact of image resolution on model learning and performance, comparing CNNs trained at five image sizes ($64 \times 64$ to $1024 \times 1024$). With comparable AUC values obtained at different resolutions, our experiments using three activation mapping techniques (saliency, Grad-CAM, XRAI) revealed considerable in-lung attention growth with increased resolution. The highest resolution models focus attention on the lungs, necessary for radiographic diagnosis of pulmonary edema. Our results emphasize the need to utilize radiographs of near-native resolution for optimal CNN performance, not fully captured by summary metrics like AUC.

**Keywords:** explainability, pulmonary edema, radiograph, resolution, visualization.

## 1. Introduction

Pulmonary edema is a condition characterized by excess fluid in the lungs, often caused by congestive heart failure (HF), among other etiologies (Staub, 1974; Murray, 2011). Due to its wide availability and ability to provide alternative diagnoses with similar features, chest radiographs are commonly used to monitor the progression of pulmonary edema (Hammon et al., 2014; Halperin et al., 1985). However, radiographic assessment of pulmonary edema is a challenging visual task, especially in mild and moderate cases, even for expert subspecialty

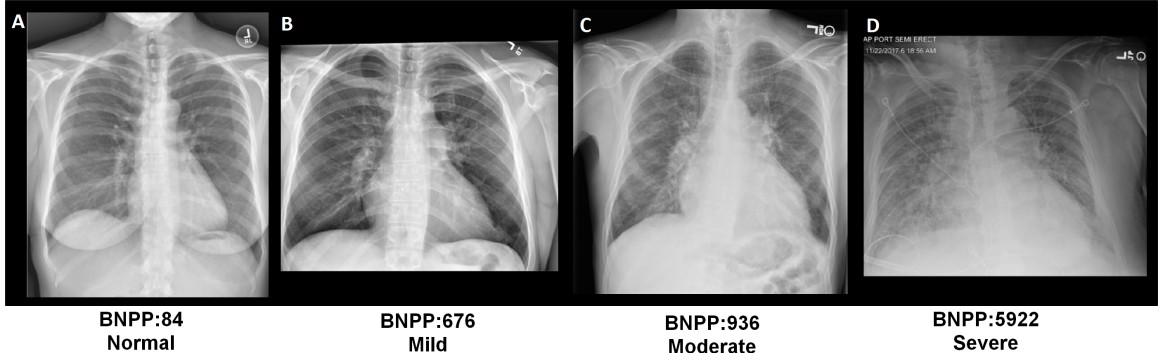

Figure 1: Relationship of radiographic appearance to BNPP and radiologist grade of severity of pulmonary edema.

cardiothoracic radiologists. Accurate assessment of pulmonary edema is crucial for guiding and monitoring response to treatment.

Recently, several groups (Lakhani and Sundaram, 2017; Wang et al., 2020; Hurt et al., 2020a,b; Hwang et al., 2019; Rajpurkar et al., 2018) have reported the application of deep convolutional neural networks (CNNs) to classify chest radiographs for various pathologies, including pneumonia, pulmonary edema, pneumothorax, and many others. While these early works show the promise of CNNs for radiographic interpretation, most lack the specificity and granularity in diagnosis at a level that is typically required for diagnostic utility. One obstacle that impedes the development of CNNs for analysis of medical images is the need to assemble ground truth data based on expert opinion. This can be labor- and time-intensive, and for challenging tasks like assessment of pulmonary edema, can be difficult to ensure reliability of image annotation. Herein, we explored the potential to infer B-type natriuretic peptide (BNP) and NT-pro B-type natriuretic peptide (BNPP) from chest radiographs (Figure 1), proposing an *objective* source of ground truth for training CNNs to perceive variations in severity of pulmonary edema. BNP and BNPP are biomarkers measured from blood serum and may be included in the diagnostic workup of suspected cardiogenic pulmonary edema (Ware and Matthay, 2000). Elevated values of BNP and BNPP are indicative of atrial stretch, observed in acute heart failure and pulmonary edema (Huang et al., 2016; Ray et al., 2005).

We further observed that in the published literature, many CNN algorithms have been trained and evaluated on low-resolution images, commonly provided in public databases (Pan et al., 2019; Jaeger et al., 2014; Seah et al., 2019). However, many of the characteristics of pulmonary edema lie near the native resolution of chest radiographs, including interstitial Kerley B lines and peribronchial cuffing. In this work, we investigated the ability of CNNs to infer BNP/BNPP from pulmonary edema radiographics when trained using different image sizes ($64 \times 64 - 1024 \times 1024$).

## 2. Methods

### 2.1. Dataset

With institutional review board approval and waiver of informed consent, we constructed a dataset of 27748 frontal chest radiographs with BNP or BNPP laboratory values from 16401

Table 1: Data Used for Train, Validation, and Test, with no Significant Difference in BNPP/BNP Value Distributions ($P > 0.1$, *Kolmogorov Smirnov Test*).

| Dataset | Description | Train 80% | Validation 10% | Test 10% |
|---|---|---|---|---|
| BNPP | Patients (n = 15409) | 12327 | 1541 | 1541 |
| | Radiographs (n = 26667) | 21374 | 2602 | 2691 |
| | Radiographs with BNPP > 400 (n = 22021) | 17631 | 2168 | 2222 |
| | Measured BNPP Value (Mean) | 4997 | 4825 | 4227 |
| | Measured BNPP Value (SD) | 11443 | 11369 | 9914 |
| BNP | Patients (n = 1325) | 1044 | 141 | 140 |
| | Radiographs (n = 1423) | 1124 | 148 | 151 |
| | Radiographs with BNP > 100 (n = 640) | 512 | 61 | 67 |
| | Measured BNP Value (Mean) | 542 | 695 | 672 |
| | Measured BNP Value (SD) | 944 | 1619 | 1205 |

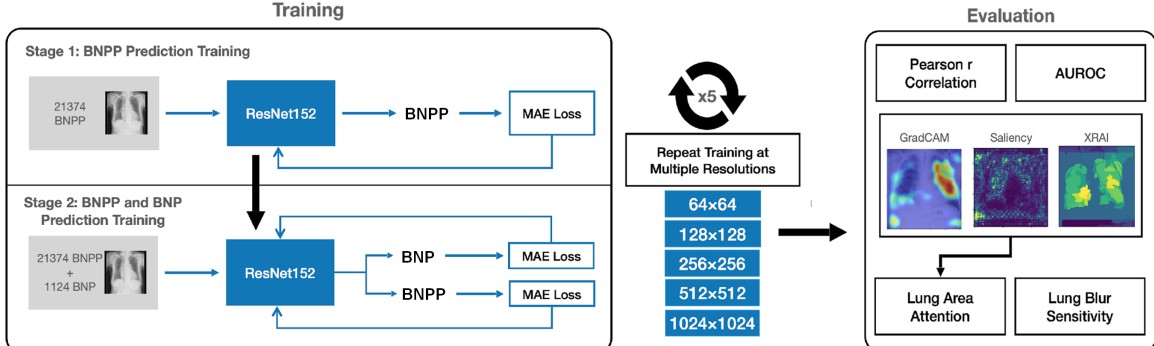

Figure 2: Flow chart of our approach to CNN training and evaluation.

patients from our institution. We included all radiographs and laboratory measurements from Nov $4^{th}$ 2014 to Dec $1^{st}$ 2020 for patients who underwent either measurement of BNP or BNPP within 24 hours of a radiograph. Depending on source x-ray device, image dimension ranges from 1400 to 4700 in height and width. The two datasets (BNPP and BNP) are described by Table 1. There was little overlap of 342 radiographs with both BNP and BNPP measurements available. Each dataset was then divided by patient, not by radiographic image, into training (80%), validation (10%) and test (10%) cohorts. There was no significant difference in BNPP or BNP value distributions between training, validation, and testing sets ($p > 0.1$, *Kolmogorov Smirnov Test*).

## 2.2. CNN Training

**Two-Stage Training:** A two-stage pipeline was used to train a bifurcated CNN to jointly predict BNP and BNPP, shown in Figure 2. All CNNs were trained using Adam optimizer with a fixed learning rate of 1e-5 for 50 epochs, and batch size 16. In the first stage of

training, a ResNet152v2 model (He et al., 2016), pretrained on the ImageNet dataset (Deng et al., 2009), was modified to infer BNPP from a chest radiograph. For training, a custom loss function based on mean absolute error (MAE) was used. Given a dataset of $n$ input radiographs, we defined the loss over the dataset as:

$$MAE^{BNPP} = \frac{1}{n} \sum_{k \in \{1,\ldots,n\}} AE_k{}^{BNPP}, \quad AE_k{}^{BNPP} = |ln(\frac{1 + y_k{}^{BNPP}}{1 + \hat{y}_k^{BNPP}})| \qquad (1)$$

where $y_k{}^{BNPP}$ is the lab measured BNPP value and $\hat{y}_k^{BNPP}$ is the inferred BNPP value for the $k^{th}$ input radiograph in the dataset. The BNPP values range from (0-70,000 pg/mL) and are exponentially distributed, with a small number of values significantly higher than the mean. To account for this and prevent overfitting to outliers using MAE loss, we used log transformation of the measured and inferred BNPP values when calculating $AE_k{}^{BNPP}$ (Cano-Espinosa et al., 2018).

In the second stage of training, an additional fully connected layer was incorporated at the last layer to predict both BNP and BNPP from a radiograph. Weights acquired from the first stage of training were frozen, except for the last fully connected layers. Both BNP and BNPP datasets were used to train the model at stage 2. Because the BNP dataset was significantly smaller than the BNPP dataset (n=1423 vs 26667 respectively), a scheduler was used to balance the number of BNP labeled radiographs and BNPP labeled radiographs in each minibatch of training examples. This ensures that for each epoch, the entire BNP training set was used (n=1124), while an equal number of BNPP labeled images were randomly sampled without replacement from the BNPP dataset. We further modified our custom MAE loss function from stage 1 (Equation (1)) to train both tasks simultaneously. To deal with missing values of BNP or BNPP measurements we ignored the outcome with missing measurement in the loss function using binary flags, $\alpha_k$ and $\beta_k$:

$$MAE = \frac{1}{n} \sum_{k \in \{1,\ldots,n\}} \alpha_k AE_k{}^{BNPP} + \beta_k AE_k{}^{BNP} \quad \alpha_k, \beta_k = \begin{cases} 1,1 & \text{if } y^{BNPP}, y^{BNP} \text{ available} \\ 1,0 & \text{if } y^{BNPP} \text{ available} \\ 0,1 & \text{if } y^{BNP} \text{ available} \\ 0,0 & \text{otherwise.} \end{cases}$$

$$(2)$$

**Training at Multiple Resolutions:** To explore the effect of image resolution on model performance, we trained five CNNs with similar architectures for different input resolutions with sizes of $64 \times 64$, $128 \times 128$, $256 \times 256$, $512 \times 512$, and $1024 \times 1024$. Images were cropped at their larger dimension to equal height and width and downscaled to the desired resolution with bilinear interpolation from python OpenCV 4.5.1.48 library. A single Nvidia V100 GPU was used to train lower resolution models ($64 \times 64 - 512 \times 512$) and 8 NVIDIA V100 GPUs from an NVIDIA DGX cluster running in an NGC container on the Singularity runtime environment were used to train the $1024 \times 1024$ CNN. Synchronous distributed training was performed using TensorFlow 2.1.0 with mirrored strategy.

### 2.3. CNN Evaluation

CNNs are evaluated in terms of area under the receiver operating characteristic curve (AU-ROC or AUC ROC) and Pearson r. ROC curves were computed after binary thresholding of BNP and BNPP measurements, according to previously established screening thresholds for

acute heart failure detection (greater than 400 for BNPP, greater than 100 for BNP) (Kim and Januzzi Jr, 2011).

## 2.4. CNN Activation Mapping

To assess the effect of resolution on CNN activation, we applied three activation mapping techniques (Saliency (Simonyan et al., 2014), grad-CAM (Selvaraju et al., 2017), and XRAI (Kapishnikov et al., 2019)) to each trained CNN. Activation maps were generated for each radiograph in the BNPP test set (n=2691).

## 2.5. Quantitative Analysis of CNN Attention

To measure the degree of CNN attention within the lungs, we propose two metrics: lung area attention (AA) and lung blur sensitivity (BS), both

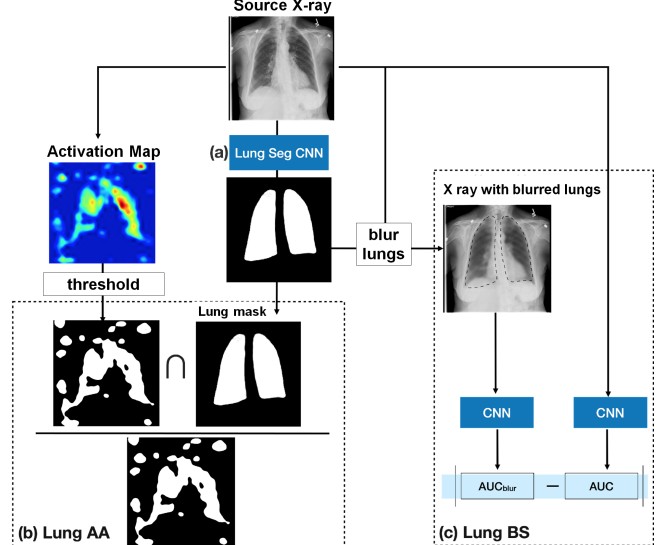

Figure 3: Proposed metrics to quantify CNN attention level in the lungs.

of which utilize lung masks from separately developed lung segmentation CNN (Figure 3-a). The lung segmentation CNN was trained using 302 radiographs and their manually-annotated lung masks, based on U-net implementation (Ronneberger et al., 2015).

**Area Attention:** We define lung area attention (AA) as the proportion of the highly activated pixels in the activation map that overlap with the lung segmentation mask (Figure 3-b):

$$AA(x) = \frac{heatmap(x) \cap mask(x)}{heatmap(x)} \tag{3}$$

where $x$ is the input chest radiograph, $heatmap(x)$ is the normalized activation map from inference on $x$, thresholded at mean pixel value across all activation maps from a single model and technique, and $mask(x)$ is the lung mask. Intuitively, a CNN with a high average lung AA value across the test set has focused mostly within the lungs rather than the rest of the image.

**Blur Sensitivity:** We define blur sensitivity (BS) as another way to estimate attention (Figure 3-c). Lung BS measures the sensitivity of the CNN to blurring the region denoted by a lung mask:

$$BS(\hat{y}, b) = AUC(\hat{y}, y) - AUC(blur(\hat{y}, b), y) \tag{4}$$

where $\hat{y}$ is a vector of the inferred values from a trained CNN for the entire test set, $blur(\hat{y}, b)$ is a vector of inferred values from when each image in the test set has lungs blurred with a gaussian kernel of size b, and $AUC(\hat{y}, y)$ is the AUC computed for a vector of inferred values $\hat{y}$ against ground truth vector $y$. We increased the Gaussian kernel sizes with respect to the image size to ensure a similar effect relative to the field of view. A model that relies on high resolution details within the lungs will have a larger lung BS value.

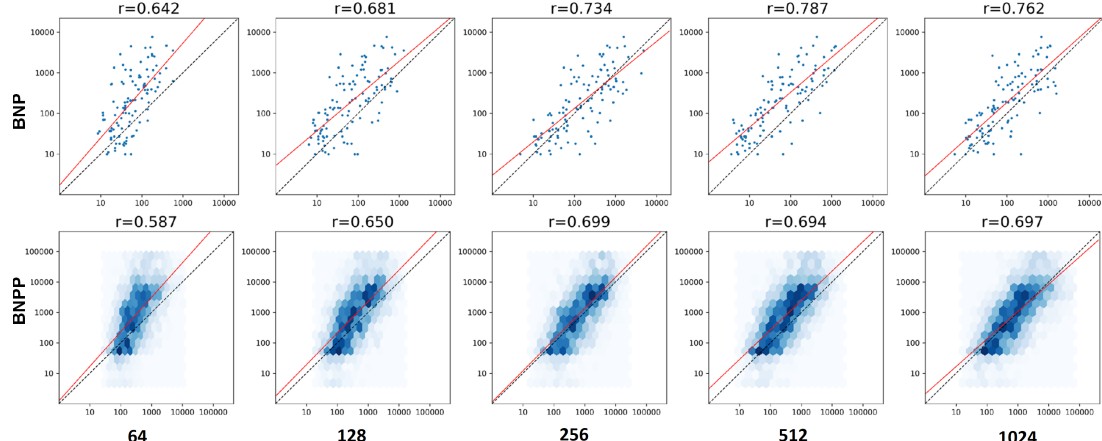

Figure 4: Measured vs inferred BNP and BNPP at multiple image resolutions.

## 3. Results

### 3.1. CNN Evaluation

The relationship between measured labora-
tory values and the respective inferred values
by CNNs are shown in Figure 4, for both BNP
and BNPP test sets at different input image
resolutions. There was relatively stronger cor-
relation in case of BNP than BNPP values at
all image resolutions, though the CNN train-
ing and evaluation sets were much smaller
(r=0.642-0.762 for BNP and r=0.587-0.697
for BNPP). Pearson correlation coefficient be-

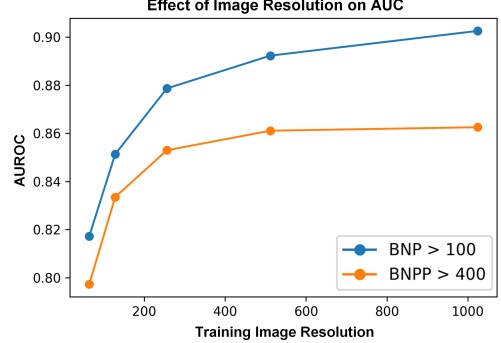

Figure 5: AUC vs input resolution.

tween measured and inferred laboratory values increased with input image resolution, having
the greatest effect at lower image resolutions. For BNP, peak Pearson r was 0.787 at 512
image size and decreased slightly to 0.762 at 1024. For BNPP inference, peak Pearson r
was 0.699 at 256 which plateaued at higher image sizes.

The relationship between input image resolution and AUC obtained for BNP and BNPP
prediction, thresholded at 100 and 400 respectively for acute heart failure detection are
shown in Figure 5. Increasing the image size from 64 to 1024 resulted in continuous incre-
ments in AUC (0.817 to 0.903 for BNP and 0.797 to 0.863 for BNPP) with the greatest
improvement between the lowest resolutions. Using the Youden's j index on each AUROC
at image sizes (64-1024), we measured the sensitivity for BNP (0.618-0.882) and BNPP
(0.728–0.815) as well as specificity for BNP(0.904–0.810) and BNPP (0.728–0.723).

### 3.2. CNN Activation Mapping

**Averaged Activation Maps:** Three strategies of saliency, grad-CAM, and XRAI were
then used on CNNs trained at all image sizes to assess CNN activation maps for consistent
trends. The resulted maps at each resolution were averaged over all test cases for each
strategy, presented as average activation maps in Figure 6-A. Overall, average heatmaps
from all three techniques show increasing attention on the lungs and decreased attention

outside of the lungs with greater image resolution. For the $64 \times 64$ model, saliency and XRAI show activations throughout the entire image, while grad-CAM activations are focused on the lower right corner of the image. For the $128 \times 128$ and $256 \times 256$ models, all three techniques show increased activations concentrated in the general lung area. Saliency and grad-CAM activation maps still focus on a single large region with no distinction between left and right lungs.For the $512 \times 512$ and $1024 \times 1024$ models, all three techniques show activation in two distinct regions: the left lung and right lung, with minimal activations outside of the lungs.

### 3.3. Quantitative Analysis of Model Attention

**Area Attention:** We calculated the average lung AA over all images in the test set (n=2691) for five CNNs, trained at different input resolutions, using three activation mapping techniques (saliency, grad-CAM, XRAI) ( Figure 6-B). Overall, increasing input resolution led to increasing average lung AA (0.40 to 0.64 for saliency, 0.26 to 0.80 for grad-CAM, 0.33 to 0.72 for XRAI). The greatest changes in average lung AA were observed when input resolution was increased from $512 \times 512$ to $1024 \times 1024$ (0.46 to 0.64 for Saliency, 0.58 to 0.80 for grad-CAM, and 0.54 to 0.72 for XRAI). At $64 \times 64$ image size, average lung AA$< 0.5$, indicates that the CNN trained at this resolution, focused less than half of its attention within the lungs. In contrast, all techniques yielded average lung AA $> 0.5$ for $1024 \times 1024$ image size, indicating such model mostly focused inside the lungs. Our analyses also suggested that average lung AA is independent of BNP or BNPP values (Appendix E). Lung attention seems to be consistent regardless of BNPP, but varies greatly with input resolution.

**Blur Study:** We calculated the lung BS based on AUC obtained from the test set (n=2691). Figure 6-C plots the average lung BS for five CNNs, trained at different image resolutions. Overall, increasing input resolution resulted in lung BS increasing from 0.01 to 0.13. For the models trained at lower image resolutions ($64 \times 64 - 256 \times 256$), lung BS $< 0.02$ indicates that blurring the lungs caused trivial changes in AUC. The higher resolution models trained at $512 \times 512$ and $1024 \times 1024$ exhibit significantly higher lung BS values of 0.06 and 0.13 respectively.

### 4. Discussion

In this work, we demonstrated the feasibility of inferring BNP and BNPP from chest radiographs. A modified Resnet152V2 CNN was developed using staged training to deal with multiple data sets of different sizes. An optimal performance at larger input image size was achieved, which highlights the importance of spatial details for inferring BNP and BNPP values. At $1024 \times 1024$ image size, thresholding the inferred values at $BNP > 100$ and $BNPP > 400$, AUROCs were 0.903 and 0.863, while Pearson r values were 0.762 and 0.697. By applying three activation mapping techniques (saliency, grad-CAM, XRAI) and two proposed quantitative metrics (lung AA, lung BS) to our CNNs, we confirmed that increasing input resolution increased model attention to the lungs, the most clinically relevant region of the radiograph. To have generalized observations, we employed ResNet152v2 model architecture with minimal modifications. We chose this architecture due to its su-

perior performance in our preliminary experiments (Appendix B). Few prior investigators have begun to explore the application of CNNs to infer blood serum biomarkers from chest radiographs. As detecting pulmonary edema on chest radiograph is challenging even for expert radiologist, assembling a dataset based on expert opinion may be inconsistent, as well as time- and labor-intensive. This work utilizes serum biomarkers as objective data to drive neural network training.

(Seah et al., 2019) showed initial feasibility of using BNP for this task at $128 \times 128$ resolution which resulted in an AUROC of 0.82 on their test set, compared to our result of 0.903 AUC. Unlike our proposed model, their model attention was predominantly outside of the lungs. Other investigators who also developed other tools for detecting pulmonary edema from chest radiographs, achieved similar AUROC, ranging from 0.814-0.924 (Rajpurkar et al., 2018; Cicero et al., 2017; Sabottke and Spieler, 2020) with a variety of CNN architectures.

We thus expanded on these works and show that while some performance (in terms of AUROC) is maintained at lower image sizes, CNNs require higher resolution to ensure that their inferences are the result of lung attention. Our results provide more insight into the effect of image resolution on CNN learning. Future work can focus on developing novel architectures for this task, or on relating the BNP and BNPP values to radiologist grades of pulmonary edema.

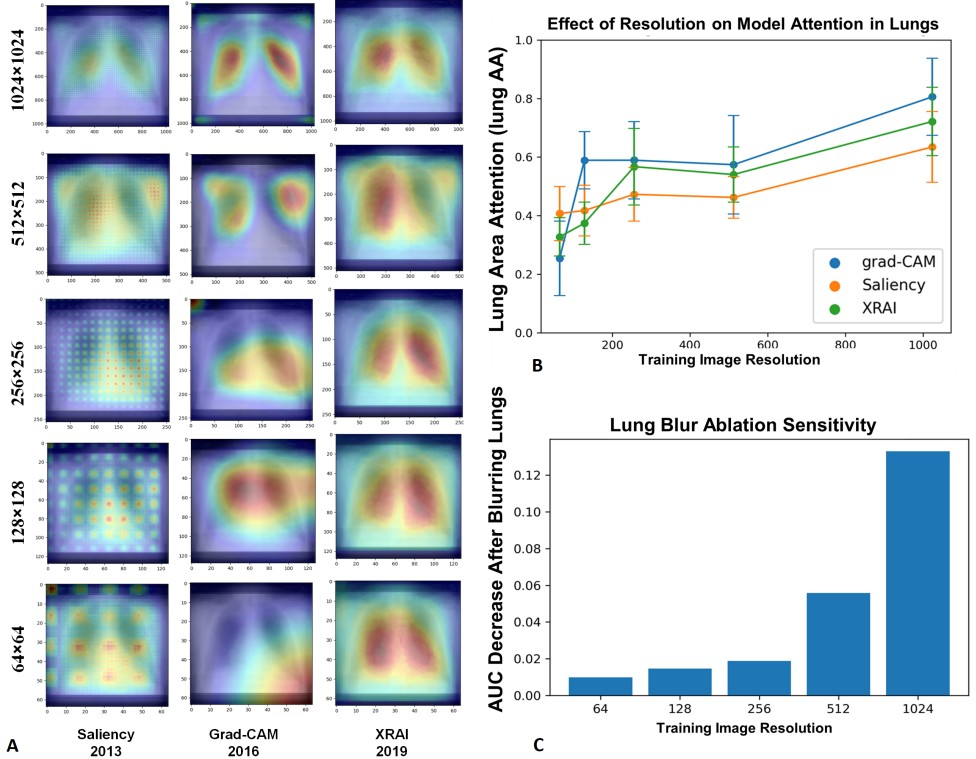

Figure 6: A) Average heatmaps from 3 visualization techniques applied to models trained at different resolutions. Image resolution vs CNN attention in the lungs using: B) lung area attention (LAA), C) lung blur sensitivity (LBS).

## Acknowledgments

This work was supported in part by the National Science Foundation under Grant IIS 2026809 and DARPA N00173-21-Q-0141. The authors acknowledge in-kind support from Microsoft AI for Health, NVIDIA and Amazon Web Services for this work. This work was completed in part at the SDSC GPU Hackathon, using resources of the Oak Ridge Leadership Computing Facility at the Oak Ridge National Laboratory. The authors would like to acknowledge OpenACC-Standard.org and NVIDIA for their support.

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

## Appendix A. Grad-CAM Visualization:

Grad-CAM activation maps were generated to visually assess the effect of resolution on CNN activation for each individual radiograph. Exemplar radiographs from two patients with varying severity of pulmonary edema are shown in Figure 7 and Figure 8. In both examples, Grad-CAM shows diffuse and inconsistent activation at lower 64 and 128 image sizes, which increasingly focus on the lungs at higher 512 and 1024 image sizes.

## Appendix B. Comparison of Resnet152V2 to other methods:

It must be noted that one of the main focuses of this work was assessing the effect of input resolution on CNN performance and attention. We wanted to ensure that the results we observed were generalizable and not the result of a specific architectural modification or technique. To have generalized observations, we employed ResNet152v2 model architecture with minimal modifications. We chose ResNet152v2 with MAE loss for its superior performance over other architectures for BNPP inference from radiographs ( Figure 9-A). This figure shows the ROC and AUC results using different methods with a threshold of 300 for measured BNPP and input image size of $512 \times 512$. Future work may focus on developing models structurally optimized for this task.

## Appendix C. ROC results using various thresholds on measured BNPP:

While in the manuscript we used a previously established screening threshold of 400 to detect acute heart failure from measured BNPP values, here we provided the ROC curves and their respective AUC computed for other potential thresholds in Figure 9-B. We also added confusion matrices calculated based on Youden's j index applied to AUROC (with $BNP_{measured} = 100$ and $BNPP_{measured} = 400$) for each model evaluated on the test set. The optimum thresholds applied to inferred values are listed in Table 2.

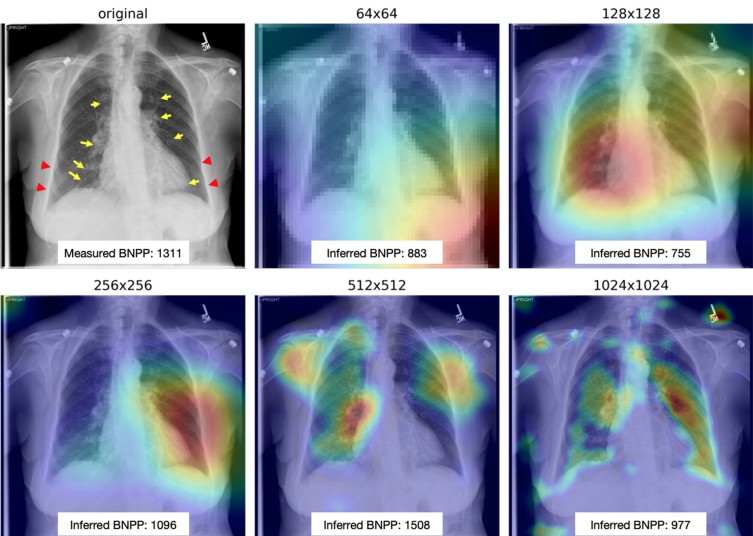

Figure 7: **Comparison of grad-CAM heatmaps from models trained at different input resolutions performing inference on a case of mild pulmonary edema**. The original image was independently annotated by cardiothoracic radiologist for peribronchial cuffing (yellow arrow) and Kerley B lines (red arrow), findings of mild pulmonary edema. Low resolution models (64, 128) show attention in large, indistinct regions on the chest X-ray. Higher resolution models (512, 1024) show greater attention in the lung regions identified by radiologist annotations.

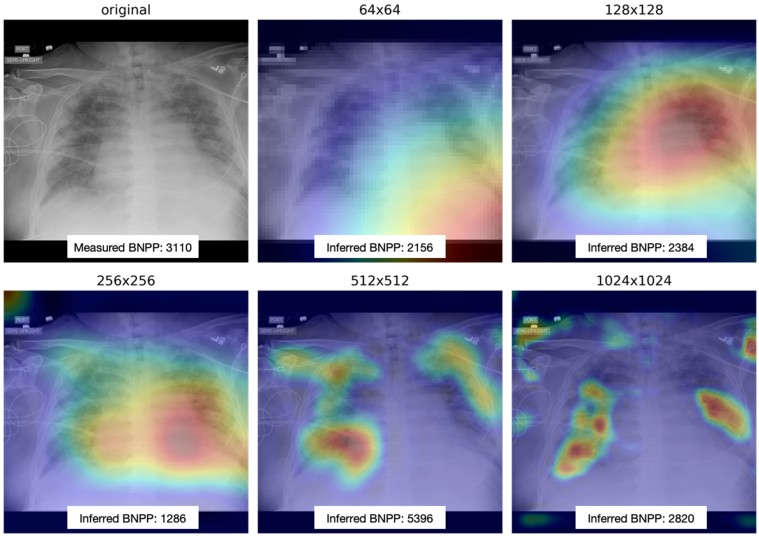

Figure 8: **Comparison of grad-CAM heatmaps from CNNs trained at different training image resolutions performing inference on a case of severe pulmonary edema**. Low resolution models (64, 128, 256) show attention in large, indistinct regions of the chest X-ray. Higher resolution models (512, 1024) show attention in the areas of alveolar opacities, the hallmark finding of severe pulmonary edema

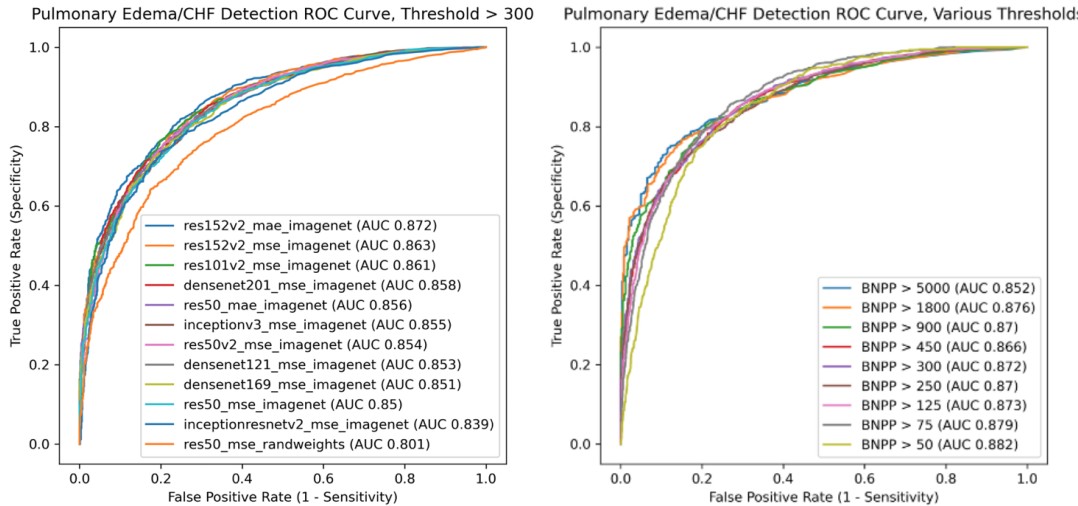

Figure 9: A) Comparison of AUROC on BNPP > 300 inference between different CNN architectures, weight initialization strategies, and loss functions when using $512 \times 512$ image size. ResNet152v2 with MAE loss was selected for subsequent experiments, B) Comparison of CNN AUROC on various BNPP thresholds, when using ResNet152v2 model and $512 \times 512$ image size.

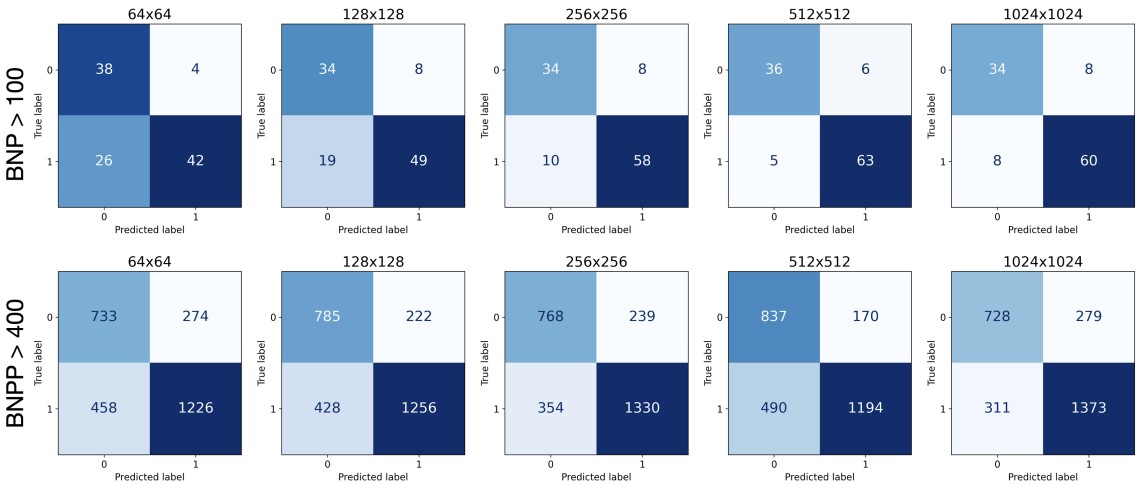

Figure 10: Confusion matrices for BNP and BNPP inference on the test sets. BNP and BNPP measured values were thresholded based on clinically established thresholds of 100 and 400 respectively. BNP and BNPP inferred values were thresholded based on Youden's J index from AUROC as in Table 2.

## Appendix D. Detailed results of BNP and BNPP prediction:

we brought together the results of our proposed method for BNP and BNPP inference from radiographs on our test set in the form of Pearson r coefficient, AUC, thresholds from AUC according to Youden's j index, sensitivity, and specificity values at various input images sizes in Table 2.

Table 2: Effect of Resolution On Pearson r Coefficient And AUC for BNP and BNPP Inference.

|  | 64 | 128 | 256 | 512 | 1024 |
|---|---|---|---|---|---|
| r (BNP) | 0.642 | 0.681 | 0.734 | 0.787 | 0.762 |
| r (BNPP) | 0.587 | 0.650 | 0.699 | 0.694 | 0.697 |
| AUC ($BNP_{measured} > 100$) | 0.817 | 0.851 | 0.879 | 0.892 | 0.903 |
| AUC ($BNPP_{measured} > 400$) | 0.797 | 0.833 | 0.853 | 0.859 | 0.863 |
| Optimum threshold ($BNP_{inferred}$) | 58 | 56 | 76 | 22 | 48 |
| Optimum threshold ($BNPP_{inferred}$) | 250 | 300 | 440 | 320 | 460 |
| Sensitivity(BNP) | 0.618 | 0.721 | 0.852 | 0.926 | 0.882 |
| Sensitivity(BNPP) | 0.728 | 0.746 | 0.790 | 0.709 | 0.815 |
| Specificity(BNP) | 0.904 | 0.810 | 0.810 | 0.857 | 0.810 |
| Specificity(BNPP) | 0.728 | 0.780 | 0.763 | 0.831 | 0.723 |

## Appendix E. Attention heatmap evaluation:

Herein, we added additional details about the feature map size used for model attention evaluation at Table 3. We also provided Table 4 which presents the average lung AA at different image resolutions. In another experiment we examined whether lung attention depends on the value of low NT-pro-BNP (Table 5). There were no significant differences in measured attention in the lungs between samples with high and low BNPP. For example, for Grad-CAM maps applied to BNPP models, the mean Lung AA for 64, 128, 256, 512 and 1024 resolution input with BNPP greater than 400 was 0.247, 0.589, 0.591, 0.566, 0.810, and for BNPP less than 400 was 0.262, 0.634, 0.593, 0.585, 0.811. Lung attention seems to be consistent regardless of BNPP, but varies greatly with input resolution.

Table 3: Effect of Resolution On Pearson r Coefficient And AUC for BNP and BNPP Inference.

|  | $64 \times 64$ | $128 \times 128$ | $256 \times 256$ | $512 \times 512$ | $1024 \times 1024$ |
|---|---|---|---|---|---|
| Feature Map Size | (2,2,2048) | (4,4,2048) | (8,8,2048) | (16,16,2048) | (32,32,2048) |

Table 4:   Effect of Resolution on Lung Area Attention.

|  | $64 \times 64$ | $128 \times 128$ | $256 \times 256$ | $512 \times 512$ | $1024 \times 1024$ |
|---|---|---|---|---|---|
| **Grad-CAM** | | | | | |
| Lung AA (Mean) | 0.259 | 0.588 | 0.587 | 0.576 | 0.803 |
| Lung AA (SD) | 0.129 | 0.096 | 0.131 | 0.172 | 0.129 |
| **Saliency** | | | | | |
| Lung AA (Mean) | 0.404 | 0.420 | 0.473 | 0.461 | 0.639 |
| Lung AA (SD) | 0.081 | 0.081 | 0.087 | 0.078 | 0.121 |
| **XRAI** | | | | | |
| Lung AA (mean) | 0.328 | 0.372 | 0.565 | 0.541 | 0.721 |
| Lung AA (sd) | 0.063 | 0.073 | 0.134 | 0.092 | 0.119 |

Table 5:   Effect of BNPP level on Lung Area Attention.

|  | $64 \times 64$ | $128 \times 128$ | $256 \times 256$ | $512 \times 512$ | $1024 \times 1024$ |
|---|---|---|---|---|---|
| **Grad-CAM** | | | | | |
| Lung AA $_{BNPP>400}$ (Mean) | 0.247 | 0.589 | 0.591 | 0.566 | 0.810 |
| Lung AA $_{BNPP>400}$ (SD) | 0.103 | 0.121 | 0.096 | 0.084 | 0.142 |
| Lung AA$_{BNPP<400}$ (Mean) | 0.262 | 0.634 | 0.593 | 0.585 | 0.811 |
| Lung AA$_{BNPP<400}$(SD) | 0.089 | 0.081 | 0.126 | 0.149 | 0.097 |

## Appendix F.  Choice of Resolution

A potential limitation of our work is that we did not experiment with resolutions higher than $1024 \times 1024$, even though the native resolution of our chest radiographs was as high as $4700 \times 4700$. For our experiments, we selected resolutions to encompass the entire gamut of commonly used input resolutions when training CNNs on chest radiographs. $1024 \times 1024$ was selected as the maximum resolution in our work for two reasons: (1) this is the maximum resolution of images from the commonly used public NIH ChestX-ray14 dataset (Jaeger et al., 2014) and RSNA-Pneumonia dataset (Pan et al., 2019). (2) Compute resources required for training increase two-fold with resolution (Table 6). Training a ResNet152v2 on $1024 \times 1024$ images was pushed the memory limits of our available hardware. Future work may be directed at studying the performance gains at even higher resolutions.

Table 6:  Effect Of Resolution on Computational cost of RESNET152V2 based Model (58M Params), Measured in Floating Point Operations (FLOPs).

|  | $64 \times 64$ | $128 \times 128$ | $256 \times 256$ | $512 \times 512$ | $1024 \times 1024$ |
|---|---|---|---|---|---|
| G-FLOPs | 0.95 | 3.62 | 14.31 | 57.07 | 228.12 |

