# OpenReview forum: "Deep Learning Radiographic Assessment of Pulmonary Edema: Training with Serum Biomarkers"
_MIDL.io/2022/Conference — MIDL 2022_

### Official Review · Reviewer_jKQT · 2022-01-12

**Confidence:** 5
**Preliminary Rating:** 5
**Recommendation:** Best Paper Award, Oral

**Summary:**

The author has focused on the blood serum biomarkers BNP and BNPP, indicative of acute heart failure (HF) and cardiogenic pulmonary
edema, paired with the chest X-ray imaging modality. They investigated the potential for inferring BNP and BNPP from chest radiographs.
This is quite interesting work in medical science.

**Strengths:**

This paper has well written. Each segment of this paper has been explained very well and references are organized very perfectly.
The author has used Train, Validation, and Test which is a very good way to validate the results.

**Weaknesses:**

Since this conference is only for medical imaging and machine learning that is more into engineering conferences. I had one concern because this paper has been written in a more engineering way and clinicians will not understand much.

**Deanonymize Review:**

yes

**Detailed Comments:**

No comments

**Final Rating After The Rebuttal:**

5: Strong Accept

**Justification Of The Final Rating:**

 This experiment has shown good results on a large dataset. I do agree to accept it now. I think this is in very good shape. The author has changed everything that the valuable content missing in the paper and whatever we were expecting to be shown.
I would strongly say to accept this paper. This is a really good approach to highlight in the scientific society.

**Paper Type:**

methodological development

**Questions To Address In The Rebuttal:**

Everything has been well written. Each segment of this paper has been explained very well and references are organized very perfectly.
There is not anything to ask majorly. I would ask to see the cross-validation (5 fold), could you provide me that table of each fold?

**Special Issue:**

yes

---

### Meta-Review · Area_Chair_tLaF · 2022-02-14

**Recommendation:** Accept (Poster)
**Confidence:** 4

**Metareview:**

The authors proposed an approach to assist the diagnosis of pulmonary oedema from X-rays by training CNNs to predict blood serum biomarkers instead of manually defined labels. The authors target what appears to be an important clinical application. The paper is clearly written. Even though not very innovative the method seems sound. The evaluation is thorough and original, and relies on a large data set.
The authors took the opportunity of the rebuttal phase to clarify a few points, both by replying to the reviewers and by uploading a revised manuscript.

----

Minor detail: the caption of Table 3 in the appendices should be corrected as it does not correspond to the content of the table.

---

### Decision · Program_Chairs · 2022-02-28

Accept